# Rural-urban disparities in full antenatal care utilization among women in Ethiopia: A further analysis of Mini-EDHS, 2019

Elsabeth Addisu[1]*, Niguss Cherie[1], Tesfaye Birhane[1], Zinet Abegaz[1], Abel Endawkie[2], Anissa Mohammed[2], Dagnachew Melak[2], Fekade Demeke Bayou[2], Ahmed Hussien Asfaw[3], Husniya Yasin[2], Aregash Abebayehu Zerga[3], Birhanu Wagaye[3], Fanos Yeshanew Ayele[3], Natnael Kebede[4], Asnakew Molla Mekonen[5], Mengistu Mera Mihiretu[5], Amare Muche[2], Yawkal Tsega[5]

1 Department of Reproductive and Family Health, School of Public Health, College of Medicine and Health Sciences, Wollo University, Dessie, Ethiopia, 2 Department of Epidemiology and Biostatistics, School of Public Health, College of Medicine and Health Sciences, Wollo University, Dessie, Ethiopia, 3 Department of Public Health Nutrition, College of Medicine and Health Sciences, Wollo University, Dessie, Ethiopia, 4 Department of Health Promotion, School of Public Health, College of Medicine and Health Sciences, Wollo University, Dessie, Ethiopia, 5 Department of Health System and Management, School of Public Health, College of Medicine and Health Sciences, Wollo University, Dessie, Ethiopia

* elsabethko@gmail.com

**Data Availability Statement:** All relevant data are within the paper and its Supporting information files.

## Abstract

### Background

Full antenatal care utilization is a key intervention that creates the opportunity to provide all the necessary health services during pregnancy that aims to reduce maternal and newborn morbidity and mortality. However, there is still a gap in utilizing this service between rural and urban women. So, this study aimed to identify the sources of variations in full antenatal care utilization between the rural and urban areas of Ethiopia.

### Methods

The study used the data on a nationwide representative sample of the Mini- Demographic and Health Survey (DHS) of Ethiopia. The data were collected from March 21, 2019, to June 28, 2019, in all regions of Ethiopia. Two stage cluster sampling techniques were used to select the study participants. This study included about 3,927 (weighted samples) of women aged from 15 to 49 years. A multivariate decomposition analysis technique was performed to observe the rural-urban disparities in full antenatal care utilization explained by residence difference in components of endowments and coefficients.

### Results

The prevalence of full antenatal care utilization was 43.25% (95% CI: 41.7%, 44.8%). The difference in the prevalence of full antenatal care utilization between rural and urban women was (rural prevalence was 27.73%, while in urban areas it was 15.52%). These results showed a statistically significant full antenatal care utilization gap in rural urban resident

**Funding:** The authors received no specific funding for this work.

**Competing interests:** All the authors have declared that no competing interests exist.

**Abbreviations:** CSA, Central Statistical Agency; EA, Enumeration Area; FANC, Full Antenatal Care; ICF, Inner City Fund; IFA, Iron Folic Acid; MEDHS, Mini Ethiopian Demographic Health Survey; PHC, Primary Health Care; WHO, World Health Organization.

women (-0.21807, 95% CI:(-0.27397, -0.16217)). The majority of the gap was explained by the covariate distribution, which accounted for 76.84%, and the rest, 23.16%, was due to the effect of covariate differences. Educational status, wealth status, religion, region, birth order, and parity differences between urban and rural women explain most of the full antenatal care utilization disparities.

## Conclusion and recommendations

There is a significant full antenatal care utilization disparity between rural and urban women in Ethiopia. This variation in the rural-urban full antenatal care utilization was explained by differences in characteristics (endowment). So to decrease this gap, emphasis should be given to resource distribution targeting rural households, improvement of maternal education and creating a platform to access information about the service and its relevance.

## Introduction

Full antenatal care (ANC) is defined as 4 or more antenatal visits, at least 1 tetanus toxoid (TT) vaccination and consumption of iron folic acid (IFA) for a minimum of 100 days [1]. The World Health Organization (WHO) aims that every pregnant woman and newborn gets quality care during pregnancy, childbirth and postnatal period in the world [2]. In 2015, nearly 303,000 women and adolescent girls died due to pregnancy and childbirth related complications [3]. At the same time, 99% of maternal deaths and 2.6 million still births occur in low-resource settings [4].

ANC reduces maternal and perinatal morbidity and mortality both directly, through detection and treatment of pregnancy-related complications, and indirectly, through the identification of women and girls at increased risk of developing complications during labor and delivery, thus ensuring referral to an appropriate level of care [5].

Indirect causes of maternal morbidity and mortality, such as HIV and malaria infections, contribute to approximately 25% of maternal deaths and near-misses, which can be easily modifiable through appropriate ANC follow-up [6].

A cohort study in Ethiopia, reported that having four or more ANC visits was significantly associated with 81.2%, 61.3%, 52.4% and 46.5%, reduction in postpartum hemorrhage, early neonatal death, preterm labor and low-birth weight, respectively [7]. A study from 27 selected countries in Sub-Saharan Africa showed that 12.2%, difference between urban and rural areas in the use of antenatal care [8]. According to the EDHS 2016 report, women having 4+ANC visits were 62.7%, and 27.3%, in urban and rural areas respectively [9].

Full antenatal care utilization has been associated with different factors. Those include women's education [8, 10–15], mass media exposure [8, 10, 13], household wealth index [8, 10, 13–15] residence [11, 14], unintended pregnancy [11, 12] and perceived the right initiation time [11, 12], long distance from health facility [10, 16], living in city administration, community women literacy, and Pastoralist region [10].

Several governmental and non-governmental organizations have made tremendous effort to improve ANC utilization. The Ethiopian government in its Health Sector Transformation Plan (2015/16–2019/20) aims to reduce maternal mortality to 199/100,000 live births and one of the strategies is achieving 95% ANC utilization of at least 4 visits [17]. Sustainable Development Goal 3.1 sets a specific target of MMR reduction below 70 by 2030 in the world [18].

However, in 2019, only 43% of women had four or more ANC visits for their most recent live birth, having a difference in rural and urban areas. Hence, this shows that it is lagging from the national target.

The applying decomposition method helps to decompose inequalities in full antenatal care utilization in urban-rural settings. Even though evidence showed a difference in service uptake and the magnitude of full antenatal care utilization and its determinants were well investigated, further variation in residence difference in full antenatal care utilization is explained has not rigorously explained using the decomposition analysis yet. In order to design appropriate intervention strategies and programs which minimize the gap between rural and urban areas, appreciating the contributing factors in the rural-urban differences of full antenatal care utilization has great importance. Hence, this study aimed to identify the source of variation in full antenatal care utilization between the rural and urban areas in Ethiopia by employing multivariate decomposition analysis technique.

## Methods and materials

### Source of data and populations

This study was based on the 2019 Mini-Ethiopian Demographic Health Survey (MEDHS) dataset. The data were obtained by sending a request letter of DHS program. The full dataset in STATA file format was downloaded from MEASURE DHS website: https://dhsprogram.com after receiving the authorization/permission letter to access the dataset from the CSA DHS program. The data were collected from March 21, 2019, to June 28, 2019, in all regions of Ethiopia [19].

The source population for this study was all women (15–49 years) who had a live birth in the five years preceding the survey residing in the selected clusters of, 2019 Mini-Ethiopian Demographic and Health Survey. The sampled population was all reproductive-age women in each household in the enumeration area, and focused on their pregnancies during the 5 years preceding the survey.

### Sample size

This study included 3979 women aged 15–49 years having lived birth five years prior to the survey.

### Sampling procedure

The 2019 MEDHS sample was stratified and selected in two stages. Each region was stratified into urban and rural areas, yielding 21 sampling strata. Samples of enumeration areas (EAs) were selected independently in each stratum in two stages. In the first stage, a total of 305 EAs (93 from urban and 212 from rural areas) were selected proportional to the size of the EA (based on the 2019 PHC frame) and with independent selection in each sampling stratum. A household listing operation was carried out in all selected EAs from January to April 2019. The resulting lists of households served as a sampling frame for the selection of households in the second stage. Some of the EAs selected for the 2019 MEDHS were large, with more than 300 households. To minimize the task of household listing, each large EA selected for the 2019 MEDHS was segmented. Only one segment was selected for the survey, with probability proportional to the segment size. Household listing was conducted only in the selected segment. In the second stage of selection, a fixed number of 30 households per cluster were selected with an equal probability through systematic selection from the newly created household listing. All

women aged 15-49years who were either permanent residents or visitors and slept in the household the night before the survey were eligible for an interview [19].

## Study variable measurement

**Dependent variable.** Full antenatal care utilization was the outcome of interest and was assessed using self-reported data on pregnancy care of all live births that happened within 5 years of the dates of the surveys. For analysis, it was measured as a binary variable and was categorized as "yes" if a woman had at least four ANC visit and less than 4 ANC visits as "no"

**Equity stratifier variable.** Place of residence of the woman was the key grouping independent variable which has a binary outcome categorized into "urban" coded as 0 and "rural" coded as 1.

**Independent variables.** Socio-demographic characteristics: Women's age in years (15–24, 25–34 or 35–49), Women's education (no education, primary education, secondary education or higher education), Marital status (currently married or currently not married), religion (Orthodox, Muslim, protestant or others), wealth status (poor, middle or rich), region (pastoralist, urban, or agrarian), parity/number of living children (1 child, 2–3 children or 4+ children), and birth order (first, 2-3rd, 4-5th or 6+)

## Operational definitions

**Regions.** Agrarian region(Amhara, Harari, Oromia, SNNP and Tigray) whose livelihood is mainly based on agriculture considered and with better distribution of health facilities, pastoralist or emerging regions, whose livelihood are based on mainly nomadism (Somali, Benishangul-Gumuz, Gambella and Afar) were with less access of healthcare services and urban regions, those livelihoods based on employment and trade (Addis Ababa and Dire Dawa) [20, 21].

**Wealth status.** Based on the number and kinds of consumer goods they own, households are given scores. Principal component analysis was used to derive these scores. National wealth quintiles are compiled by assigning the household score to each usual household member, and then dividing the distribution into five equal categories, each comprising 20% of the population from the lowest poorest to richest as first: if the percentiles of wealth score was ≤ 20%, second: if the percentile was between 20.1%–40%, middle: if percentile was between 40.1%–60%, fourth: if percentile was between 60.1%–80% and highest: if the percentile was ≥80.1% wealth quintile and for the current analysis we combine poorest and poorer as poor, richest and richer as rich and middle as it is [9].

## Data processing and analysis

Important variables were extracted from the data set. Data management and statistical analyses were conducted using STATA/MP 17.0 software. Weighted frequencies and percentages were calculated to account design effects. After the data were cleaned and weighted descriptive statistics were reported as means with standard deviation (SD), percentage, frequency, and tables. Multivariate decomposition analysis technique was performed. The multivariate decomposition technique enabled us to see the determinants of the differences in an outcome (full ANC) for two groups (rural and urban). The disparity in full ANC for rural and urban groups can be explained by differences in the level or distribution of the determinants of the outcome (explained component/covariates effect) and in the impacts of the determinants on the outcome (unexplained component/coefficients effect), and/or the interaction of the two components [22]. Multivariate decomposition technique determines the high outcome group automatically and uses the low outcome group as a reference category [23].

## Ethical considerations

The study used publicly available data from the 2019 Ethiopian Mini- Demographic and Health Survey. This survey was approved by Inner City Fund (ICF) international as well as the EPHI Institutional Review Board to ensure that a protocol is in compliance with the US Department of Health and Human Services regulations for the protection of human subjects. The survey data were received from the DHS International Program upon submission of a proposal. Confidentiality was maintained after data access was authorized by DHS. All methods were performed in accordance with the Declaration of Helsinki and Ethiopian research guidelines.

# Results

## Socio-demographic and maternal characteristics

A total of 3927 (weighted samples) of women were included in this study. The mean age of mothers was 28.9±0.1 years. The majority of (36.78%) of rural and (13.95%) of urban women were found in the age range of 25–34 years old. Regarding marital status (70.89%) of rural and (24.33%) urban women were currently married. Among respondents (26.61%) of rural and (10.09%) of urban residents were Orthodox Christian follower. Nearly half (43.55%) of rural and (7.75%) of urban residents have no education. Regarding the wealth status distribution, (38.81%) and (3.14%) of rural and urban residents were poor in wealth status, respectively. Sixty- eight percent (67.76%) of rural and (19.94%) of urban residents were from an agrarian region. Thirty- six percent (36.01%) of the study participants from rural and (7.36%) of urban areas have four or more live births. Regarding birth order (20.66%) of rural area and (4.20%) of urban area respondents were with a birth order of six and more children (Table 1).

**Rural-urban disparities in full antenatal care utilization among reproductive age women in Ethiopia.** The prevalence of full antenatal care utilization was 43.25% (95% CI: 41.7%, 44.8%). The disparity of full antenatal care utilization between rural and urban areas was (rural prevalence was 27.73%, while in urban areas it was 15.52%).

**Decomposition result.** The detail decomposition result showed that there is a significant disparity in full antenatal care between rural and urban residences (-0.218, p < 0.001). The difference in full antenatal care between rural and urban residencies was explained by a component of compositional differences (covariate distribution), which accounted for 76.84% and 23.16% was supplied by a component of difference in behaviour (the effect of covariate differences).

**Difference in characteristics (covariate distribution).** The difference in characteristics (endowments) accounted for 76.84% of the observed residence differences in full antenatal care uptake, with high intercept differences (-0.17, p<0.001).

The majority of the gap in full antenatal care utilization was explained by the wealth status difference between rural and urban women; both poor (30.28%) and middle wealth status (8.92%) distribution contributed for widening of the gap. Shifting rural pastoralist (-3.28%) distribution to urban levels contributed for narrowing of the gap. After controlling for other factors, the distribution of women's education with primary (3.76%), secondary (11.84%), and higher (15.27%) education levels would be expected to widen the rural-urban full antenatal care utilization disparity. That is, if urban women's distribution of primary, secondary and higher education shifted to rural levels, the rural-urban disparity would narrow by 3.76%, 11.84%, and 15.27% respectively. The distribution of religion of women, both Orthodox Christian (0.80%) and other (Catholic & traditional) (2.37%) religious followers contributed for

**Table 1. Characteristics of independent variables with full antenatal care and with stratifier variables of study participants using 2019 MEDHS, 2023; Weighted n = 3927, unweighted n = 3979.**

| Variables | Variable Categories | Full antenatal care visit | | Residence | |
|---|---|---|---|---|---|
| | | Yes(1) | No(0) | Rural(1) | Urban(0) |
| Age in years | 15–24 | 419(10.66%) | 577(14.70%) | 719(18.31%) | 277(7.06%) |
| | 25–34 | 926(23.59%) | 1066(27.14%) | 1444(36.78%) | 548(13.95%) |
| | 35–49 | 354(9.00%) | 585(14.90%) | 737(18.77%) | 202(5.13%) |
| Marital status | Currently Married | 1637(41.70%) | 2102(53.53%) | 2784(70.89%) | 956(24.33%) |
| | Currently not married | 61(1.55%) | 127(3.22%) | 116(2.97%) | 71(1.81%) |
| Religion | Orthodox Christian | 759(19.33%) | 682(17.37%) | 1045(26.61%) | 396(10.09%) |
| | Muslim | 505(12.86%) | 835(21.26%) | 1036(26.38%) | 304(7.73%) |
| | Protestant | 424(10.80%) | 658(16.77%) | 760(19.36%) | 323(8.21%) |
| | Others* | 11(0.27%) | 53(1.35%) | 59(1.51%) | 4(0.11%) |
| Women education | No education | 657(16.74%) | 1357(34.56%) | 1710(43.55%) | 305(7.75%) |
| | Primary | 669(17.02%) | 746(19.01%) | 991(25.23%) | 424(10.80%) |
| | Secondary | 252(6.41%) | 93(2.36%) | 165(4.21%) | 179(4.56%) |
| | Higher | 121(3.08%) | 32(0.81) | 34(0.87%) | 119(3.02%) |
| Wealth status | Poor | 477(12.15%) | 1170(29.79%) | 1524(38.81%) | 124(3.14%) |
| | Middle | 297(7.57%) | 465(11.82%) | 701(17.86%) | 60(1.53%) |
| | Rich | 924(23.53%) | 594(15.13%) | 675(17.19%) | 843(21.47%) |
| Parity | 1 | 450(11.45%) | 436(11.11%) | 602 (15.33%) | 284(7.22%) |
| | 2–3 | 638(16.25%) | 700(17.83%) | 884(22.51%) | 454(11.56%) |
| | 4+ | 611(15.56%) | 1092(27.81%) | 1414(36.01%) | 289(7.36%) |
| Birth order | First child | 427(10.85%) | 399(10.16%) | 545(13.87%) | 281(7.14%) |
| | 2–3 children | 615(15.66%) | 661(16.84%) | 832(21.19%) | 444(11.31%) |
| | 4–5 children | 354(9.01%) | 495(12.61%) | 712(18.14%) | 137(3.49%) |
| | 6+ children | 303(7.72%) | 673(17.13%) | 811(20.66%) | 165(4.20%) |
| Residence | Urban | 610(15.52) | 417(10.62) | - | - |
| | Rural | 1089(27.73) | 1811(46.13) | - | - |
| Region | Pastoralist | 76(1.95%) | 259(6.59%) | 229(5.84%) | 107 (2.71%) |
| | Urban | 118(3.01%) | 30(0.75%) | 11(0.27%) | 137 (3.49%) |
| | Agrarian | 1504(38.30%) | 1940(49.40%) | 2660(67.76%) | 783(19.94%) |

*- Catholic and traditional religion follower

widening of the gap. That is, if rural women distribution of orthodox Christian and Catholic & traditional religious followers shifted to urban levels, the rural-urban disparity would decrease by 0.80% and 2.37% respectively.

**Difference due to coefficients (the effect of covariate differences).** Differences in effects (difference due to coefficients) account for 23.16% of the observed rural- urban disparity in full antenatal care utilization.

The effect of Catholic and traditional (1.13%) religious followers contribute to widening the gap in full antenatal care utilization between rural and urban women. The effect of having one living child (para one) (69.29%) contributes to widening the gap in full antenatal care utilization between rural and urban women. Whereas, the effect of birth order of the first child (-68.68%) contributes to narrowing the gap in full antenatal care utilization between rural and urban women if the urban women's birth order shifts to rural level (Table 2).

**Table 2. Detailed decomposition of full antenatal care utilization by place of residence among reproductive age women using 2019 MEDHS, 2023.**

| Decomposition | Coefficient with 95% CI | | | | Percent | P-value |
|---|---|---|---|---|---|---|
| Raw Difference | -0.21807 (-0.27397, -0.16217) | | | | 100 | 0.001 |
| Explained(E) | -0.16756 (-0.20616, -0.12896) | | | | 76.84 | 0.001 |
| Unexplained(C) | -0.05051 (-0.11642, 0.01541) | | | | 23.16 | 0.133 |
| | Difference due to characteristics (E) | | | Difference due to coefficients (C) | | |
| | Coefficient | Percent | P–value | Coefficient | Percent | P -value |
| **Age in years** | | | | | | |
| 15–24 | 0.00219 | -1.00 | 0.053 | -0.00838 | 3.84 | 0.750 |
| 25–34 | -0.00023 | 0.10 | 0.858 | -0.00913 | 4.19 | 0.814 |
| 35–49 | 1 | | | 1 | | |
| **Marital status** | | | | | | |
| Currently Married | 0.00342 | -1.57 | 0.061 | -0.12179 | 55.85 | 0.147 |
| Currently not married | 1 | | | 1 | | |
| **Religion** | | | | | | |
| Orthodox Christian | -0.00175 | 0.80 | 0.039 | 0.01600 | -7.34 | 0.532 |
| Muslim | 1 | | | 1 | | |
| Protestant | 0.00240 | -1.10 | 0.231 | 0.03251 | -14.91 | 0.153 |
| Other* | -0.00517 | 2.37 | 0.010 | -0.00247 | 1.13 | 0.010 |
| **Women education** | | | | | | |
| No education | 1 | | | 1 | | |
| Primary | -0.00819 | 3.76 | <0.001 | 0.01237 | -5.67 | 0.643 |
| Secondary | -0.02583 | 11.84 | <0.001 | -0.01870 | 8.57 | 0.240 |
| Higher | -0.03329 | 15.27 | 0.013 | 0.00321 | -1.47 | 0.847 |
| **Region** | | | | | | |
| Pastoralist | 0.00716 | -3.28 | <0.001 | -0.00240 | 1.10 | 0.754 |
| Urban | 1 | | | 1 | | |
| Agrarian | -0.01262 | 5.79 | 0.094 | 0.03709 | -17.01 | 0.402 |
| **Wealth status** | | | | | | |
| Poor | -0.06586 | 30.28 | <0.001 | 0.00450 | -2.06 | 0.674 |
| Middle | -0.01945 | 8.92 | 0.004 | 0.00534 | -2.45 | 0.436 |
| Rich | 1 | | | 1 | | |
| **Parity** | | | | | | |
| 1 | 0.00124 | -0.57 | 0.866 | -0.015110 | 69.29 | 0.036 |
| 2–3 | 1 | | | 1 | | |
| 4+ | -0.00334 | 1.53 | 0.813 | -0.07675 | 35.19 | 0.174 |
| **Birth order** | | | | | | |
| First child | -0.00393 | 1.80 | 0.674 | 0.14978 | -68.68 | 0.038 |
| 2–3 children | 1 | | | 1 | | |
| 4–5 children | 0.00215 | -0.98 | 0.783 | 0.03368 | -15.45 | 0.184 |
| 6+ children | -0.00646 | 2.96 | 0.496 | 0.03261 | -14.95 | 0.344 |

*- Catholic and traditional religion follower

## Discussion

The result of this study showed that the disparity in the prevalence of full antenatal care utilization between rural and urban areas was high. More than three-fourths of the observed rural-urban full antenatal care utilization disparities among women could be attributed to

differences in composition. This implies that the rural-urban gap in full antenatal care would be reduced more by changes in characteristics (endowment) than by change in behaviour (coefficient). This finding is supported by previous studies conducted in Ethiopia [14, 24], Sub-Saharan Africa [8], and Vietnam [25], which found a significant difference in full antenatal care utilization between rural and urban residents. This might be because women having rich wealth status, educated mothers, and living in urban region have a better chance of utilizing optimum antenatal care.

The findings of this study revealed that women's education was an important predictor for widening the rural-urban disparities in full antenatal care utilization. Shifting rural women's educational status to that of urban women at the primary, secondary and higher educational levels would decrease the rural-urban disparity in full antenatal care uptake. This finding is supported by results in Ethiopia [12–14, 24], Sub-Saharan Africa [8] and North India [26]. The possible justification might be women, who are educated more, tend to use antenatal care, have better understanding and knowledge about importance of the service. Moreover, educated women are more likely to improve independency, self-confidence and ability to make decisions about their own health and seek out higher quality services and greater ability to use health care inputs that offer better care [13].

Shifting rural wealth status to urban levels would provide a significant contribution to decrease the gap between rural-urban full antenatal care utilization. This result is in line with studies conducted in Ethiopia [13, 14, 24], Sub-Saharan Africa [8], and North India [26]. It is obvious that rich women in urban areas have easy access to information regarding maternal and child health through different media, which helps them to have ANC follow up. However, rural women of poor and middle wealth status might be incapable of covering transportation and other related expenses to access the health facility, even though the ANC service is free from charge, so for to such reasons, they might not utilize the service [27].

Shifting rural distribution of both orthodox Christian and other (Catholic& traditional) religious followers to urban levels would narrow the gap between full antenatal care utilization between rural and urban women. Even though sampling might affect this result, the possible explanation might be within this setting, there may be longstanding perceptions, beliefs, and customs in the society that could lead to the underutilization of these services. [13, 28].

Our finding showed that shifting rural pastoralist distribution to urban levels would significantly contribute to the narrowing of the gap. This finding is similar to that of previous studies conducted in Ethiopia [15, 16, 24]. This could be because women residing in urban regions are capable of getting better information regarding ANC service and have better infrastructure (physical access to health facilities) than women residing in pastoralist geographic region.

The result of this study revealed that the effect of having one living child (being Para one) contributes to widening the gap in full antenatal care utilization between rural and urban women. This implies rural para one women may have limited information about this service as compared to their counterparts. However, urban women have better health-seeking behavior due to access to information and infrastructure. So the effect of shifting rural Para one women to that of urban level would be expected to narrowing the gap in full antenatal care utilization between rural and urban residents.

## Limitations and strength

One limitation is the cross-sectional nature of the study design limits the ability to establish temporal relationships. Second, both the dependent and independent variables were self-reported and are likely to have a risk of recall bias. Moreover, this analysis considers a few variables recorded in the Mini EDHS survey and there are variables not collected like that of the

main EDHS survey like media exposure, husband education, occupation, even though these are not the only factors which affect FANC utilization. Despite these limitations, this study is important in that it gives an understanding and quantification of the contributors and magnitude of full antenatal care utilization differences in rural-urban settings of Ethiopia. And this enables us to give more attention to the disadvantageous groups while planning and implementing strategic interventions targeting to the drivers of these disparities.

## Conclusion and recommendations

There was a significant rural-urban disparity in full antenatal care utilization among reproductive-age women in Ethiopia. A large portion of the rural-urban disparity in full antenatal care utilization was explained by an endowment (differences in characteristics). Women's educational status, wealth status, region, parity, birth order, and religion of women were the determinants of the rural-urban disparity in full antenatal care utilization. To narrow the rural-urban full antenatal care uptake disparity, emphasis should be given to both resource distribution targeting rural households, improving maternal education status and creating a platform to access information about the service and its relevance improving health- seeking behaviour of women.

## Supporting information

**S1 File. Mini EDHS 2019 data set in excel Stata form.**
(XLSX)

**S1 Data.**
(XLSX)

## Acknowledgments

We would like to acknowledge IPUMS DHS for providing the requested dataset.

## Author Contributions

**Conceptualization:** Elsabeth Addisu, Niguss Cherie, Aregash Abebayehu Zerga, Asnakew Molla Mekonen, Yawkal Tsega.

**Data curation:** Elsabeth Addisu, Niguss Cherie, Tesfaye Birhane, Anissa Mohammed, Dagnachew Melak, Fanos Yeshanew Ayele, Mengistu Mera Mihiretu.

**Formal analysis:** Elsabeth Addisu, Tesfaye Birhane, Abel Endawkie, Anissa Mohammed, Amare Muche.

**Investigation:** Elsabeth Addisu, Zinet Abegaz, Anissa Mohammed, Ahmed Hussien Asfaw, Birhanu Wagaye, Mengistu Mera Mihiretu, Yawkal Tsega.

**Methodology:** Elsabeth Addisu, Niguss Cherie, Zinet Abegaz, Abel Endawkie, Fekade Demeke Bayou, Husniya Yasin, Aregash Abebayehu Zerga, Natnael Kebede, Asnakew Molla Mekonen, Amare Muche, Yawkal Tsega.

**Project administration:** Elsabeth Addisu, Tesfaye Birhane, Zinet Abegaz, Abel Endawkie, Anissa Mohammed, Ahmed Hussien Asfaw, Husniya Yasin, Aregash Abebayehu Zerga, Fanos Yeshanew Ayele, Asnakew Molla Mekonen, Mengistu Mera Mihiretu, Yawkal Tsega.

**Resources:** Fekade Demeke Bayou, Birhanu Wagaye, Natnael Kebede.

**Software:** Elsabeth Addisu, Tesfaye Birhane, Zinet Abegaz, Dagnachew Melak, Fekade Demeke Bayou, Husniya Yasin, Birhanu Wagaye, Amare Muche.

**Supervision:** Dagnachew Melak, Ahmed Hussien Asfaw, Fanos Yeshanew Ayele, Natnael Kebede, Asnakew Molla Mekonen.

**Validation:** Elsabeth Addisu, Anissa Mohammed, Fanos Yeshanew Ayele.

**Visualization:** Husniya Yasin, Birhanu Wagaye, Natnael Kebede, Mengistu Mera Mihiretu, Yawkal Tsega.

**Writing – original draft:** Elsabeth Addisu, Niguss Cherie, Aregash Abebayehu Zerga.

**Writing – review & editing:** Abel Endawkie, Amare Muche, Yawkal Tsega.

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
