## [Decision Letter · Decision Letter 0]

11 Oct 2023

PONE-D-23-21294Rural-urban disparities in full antenatal care utilization among women in Ethiopia: a Multivariate decomposition analysisPLOS ONE

Dear Dr. Addisu,

Thank you for submitting your manuscript to PLOS ONE. After careful consideration, we feel that it has merit but does not fully meet PLOS ONE’s publication criteria as it currently stands. Therefore, we invite you to submit a revised version of the manuscript that addresses the points raised during the review process

We look forward to receiving your revised manuscript.

Kind regards,

Trhas Tadesse Berhe, PhD

Academic Editor

PLOS ONE

Journal Requirements:

- https://doi.org/10.1371/journal.pone.0284382

- https://doi.org/10.1186/s12884-020-03236-9

In your revision ensure you cite all your sources (including your own works), and quote or rephrase any duplicated text outside the methods section. Further consideration is dependent on these concerns being addressed.

Additional Editor Comments:

Dear Elsabeth Addisu

Following review of your article to PLOS ONE, we invite you to submit a major revision.

The review comments can be found at the end of this email, together with any comments from the Editorial Office regarding formatting changes or additional information required to meet the journal’s policies at this time.

Please note that your revision may be subject to further review and that this initial decision does not guarantee acceptance.

Editor(s)' Comments to Author (if any):

• Please ensure that your abstract adheres to our Instructions for Authors' formatting guidelines.

• Kindly include the requested supplementary file.

• Ensure consistency in the description of your study design (Is this a community-based cross-sectional study?).

• Offer clear operational definitions for "Urban" and "Rural."

• Thoroughly depict the study area, period, design, data sources, study population, sample size determination method, sampling technique, inclusion and exclusion criteria, and other pertinent details.

Best regards , Trhas Tadesse ( PhD, associate professor in public health )

Reviewer 1:

Comments to the Author

1. Is the manuscript technically sound, and do the data support the conclusions?

Response: yes

2. Has the statistical analysis been performed appropriately and rigorously?

Response: yes

3. Have the authors made all data underlying the findings in their manuscript fully available?

Response: Yes

4. Is the manuscript presented in an intelligible fashion and written in standard English?

Response : Yes

5. Review Comments to the Author

Response : What is the research's ground-breaking and original discovery? I suggest that the authors consider delving deeper into the factors contributing to disparities among rural and urban mothers. The factors mentioned and addressed by the authors are commonplace and frequently discussed.

Reviewers' comments:

Reviewer's Responses to Questions

**Comments to the Author**

1. Is the manuscript technically sound, and do the data support the conclusions?

Reviewer #1: Yes

Reviewer #2: Partly

2. Has the statistical analysis been performed appropriately and rigorously? 

Reviewer #1: Yes

Reviewer #2: N/A

3. Have the authors made all data underlying the findings in their manuscript fully available?

Reviewer #1: Yes

Reviewer #2: Yes

4. Is the manuscript presented in an intelligible fashion and written in standard English?

Reviewer #1: Yes

Reviewer #2: Yes

5. Review Comments to the Author

Reviewer #1: What is the new and novel finding of this research works/

I would recommend the authors if they go a bit deep dive to the factors contributing for disparity among rural and urban mothers. What have been mentioned and addressed by the authors are of common and most frequently addressed factors

Reviewer #2: Dear authirs,

Tye strength of your paper is

a. Addressing sensituve.issues of maternal health

b. Upto dated data of mini EDHS,2019.

3. Working together for publication

4. High level analysis

However, you have failed to achieve;

1.Depth of the issue e.g MOH ANC new guideline was not cited

2. The methods is not clear and the selection of variables need strong conviction

3. The title, the objectives and the contents are somewhat inconsistent.

4. The consistency between the title and the contents and objective and role of authors should be clearly stipulated.

5. You didin' cite the DHS in some parts and even the citation is wrong.E.g see the reference section.

E.g. the sampling procedure and definitions are already stated in EDHS itself.

6. In general, all sections need meticulous explanation and justification.

7. Try to change the majority of the article.

Regards,

6. PLOS authors have the option to publish the peer review history of their article (what does this mean?). If published, this will include your full peer review and any attached files.

Reviewer #1: **Yes: **Abebe Sorsa

Reviewer #2: No

---

## [Author Response · Author response to Decision Letter 0]

15 Nov 2023

Responses to Editor and Reviewers

Dear to Editor and Reviewers; 

We are very thankful for your invaluable comments given. Here are the authors’ responses to the comments.

We have attached the revised manuscript based on your comments.

Editor comments:

Comment 1: Please ensure that your manuscript meets plos one's style requirements, including those for file naming.

Response 1: We have taken your comments and have been revised according to the journal requirement.

Comment 2: Please ensure that you have an ORCID ID and that it is validated in editorial manager.

Response 2: I have an ORCID ID and is validated in editorial manager

Comment 3: We noticed you have some minor occurrence of overlapping text with the following previous publication(s), which needs to be addressed

Response 3: We have checked and rephrased some texts which seems having overlapping.

Comment 4: We note that you have indicated that data from this study are available upon request.

Response 4: We have attached the dataset that were analysed in this study as supplementary file in our revision.

Comment 5: please ensure that your abstract adheres to our instructions for Author's formatting guidelines.

Response 5: Thank you and your comment take into account.

Comment 6: Kindly include the requested supplementary file.

Response 6: We have attached it.

Comment 7: Ensure consistency in the description of your study design (is this a community based cross-sectional study?)

Response 7: Thank you and we have corrected like this a nationwide representative sample of the Mini Demographic and Health Survey (DHS) of Ethiopia.

Comment 8: Offer clear operational definitions for “urban” and “rural”.

Response 8: We have used place of residency as underlying stratifier variables which is predefined during the survey and in our case we were coded rural as “1” and urban as “0”.

Comment 9: Thoroughly depict the study area, period, design, data sources, study population, sample size determination method, sampling technique, inclusion and exclusion criteria, and other pertinent details.

Response 9: Thank you and we have revised all sections of methods in our manuscript. 

Reviewer1

Reviewer #1: Comment: What is the new and novel finding of this research works?

Response: Thank you and in Ethiopia research findings showed a difference in full ANC utilization between rural and urban residents. But it was not rigorously explained using the decomposition method yet and the factors were not well established for this difference to design reasonable interventions. Therefore, this study was designed to decompose the factor that significantly contributes for these rural-urban disparities in full antenatal care utilization.

Reviewer# 2: Comments:

Comment 1: Depth of the issue e.g. MOH ANC new guideline was not cited

Response 1: Thank you for reminding, but we were citing the recent WHO guideline (WHO recommendations on antenatal care for a positive pregnancy experience) which recommends eight visit, but in this study we have taken a woman having at least four ANC visit as “yes” and less than 4 ANC visits as “no”. Because our study were based on secondary data. Comment 2: The methods is not clear and the selection of variables need strong conviction

Response 2: We have taken your comment and review our method section and since we have used secondary data the selection of variables depend on the data at hand meaning that we have included those variables included in this survey, though this Mini EDHS did not include all variables as that of Main EDHS. Because as we all understand that this survey was conducted to observe the progress of health and health related conditions focused on some indicators and did not include all variables as that of Main EDHS and we include this as limitation of the study. Please see the manuscript. 

Comment 3: The title, the objectives and the contents are somewhat inconsistent 

Response 3: Thank you and we have addressed your concerns.

Comment 4: The consistency between the title and the contents and objective and role of authors should clearly stipulated

Response 4: We have taken all your comments and corrected them accordingly. But as per our understanding the role or contribution of authors have been clearly added during submission.

Comment 5: You did not cite the DHS in some parts and even the citation is wrong. E.g. see the reference section. E.g. the sampling procedure and definitions are already stated in EDHS itself.

Response 5: Thank you, but references were inserted for sampling procedure and operational definitions. 

Comment 6: In general, all sections need meticulous explanation and justification.

Response 6: Thank you for your comments and we see our manuscript thoroughly and revised accordingly. 

Comment 7: Try to change the majority of the article

Response 7: Thank you and we have revised the whole section of the manuscript.

---

## [Decision Letter · Decision Letter 1]

21 Jun 2024

PONE-D-23-21294R1Rural-urban disparities in full antenatal care utilization among women in Ethiopia: a Multivariate decomposition analysisPLOS ONE

Dear Dr. Addisu,

Thank you for submitting your manuscript to PLOS ONE. After careful consideration, we feel that it has merit but does not fully meet PLOS ONE’s publication criteria as it currently stands. Therefore, we invite you to submit a revised version of the manuscript that addresses the points raised during the review process.  

We look forward to receiving your revised manuscript.

Kind regards,

Doris Verónica Ortega-Altamirano, PhD

Academic Editor

PLOS ONE

Additional Editor Comments:

The manuscript you submit is good. However, it can improve if you take into consideration the suggestions of the reviewers. Please send the new version of the manuscript before July 5.

Reviewers' comments:

Reviewer's Responses to Questions

**Comments to the Author**

1. If the authors have adequately addressed your comments raised in a previous round of review and you feel that this manuscript is now acceptable for publication, you may indicate that here to bypass the “Comments to the Author” section, enter your conflict of interest statement in the “Confidential to Editor” section, and submit your "Accept" recommendation.

Reviewer #2: All comments have been addressed

Reviewer #3: (No Response)

Reviewer #4: All comments have been addressed

Reviewer #5: (No Response)

2. Is the manuscript technically sound, and do the data support the conclusions?

Reviewer #2: Partly

Reviewer #3: Partly

Reviewer #4: Yes

Reviewer #5: Yes

3. Has the statistical analysis been performed appropriately and rigorously? 

Reviewer #2: Yes

Reviewer #3: Yes

Reviewer #4: Yes

Reviewer #5: Yes

4. Have the authors made all data underlying the findings in their manuscript fully available?

Reviewer #2: Yes

Reviewer #3: Yes

Reviewer #4: Yes

Reviewer #5: Yes

5. Is the manuscript presented in an intelligible fashion and written in standard English?

Reviewer #2: Yes

Reviewer #3: No

Reviewer #4: No

Reviewer #5: Yes

6. Review Comments to the Author

Reviewer #2: Review Reports

Title: Rural-urban disparities in full antenatal care utilization among women in Ethiopia: a Multivariate decomposition analysis

Number: PONE-D-23-21294R1

Review Comments

We acknowledge the recipients of the point-to-point responses of the authors to our previous comments and the incorporation of the comments in to the new submitted manuscript. In addition, we notify our receipt of the tracked changes of the accepted comments. Again, we acknowledge for conducting the study by team. The following are our comments;

a. On the title: Needs reframing and make it absorbing to the reader. For instance, If I were you, I may rewrite the title as “Rural-urban disparities in full antenatal care utilization among women in Ethiopia: A further Analysis of mini-EDHS, 2019”

b. On the abstract section

• The background is incomplete

• It lacks clarity and used inappropriate words E.g., ‘to see’

• The presentation of the result is also incomplete E.g., lacks confidence interval.

c. On the methods section

• It is known that urban dwelling women have higher full ANC utilization when compared with rural dwellers due to the gains in the urban dwelling. Hence, what new findings do this study report? Why was the specific type of analysis then employed?

• What was the reason behind the selection of those women aged 15 to 49 years (3979), the number of sampled women before your analysis in the actual EDHS, and yours (3,979)?

d. On the result and the consequent sections

The rural dwellers have higher full ANC utilization that the urban one. Access, sampling?

Try to address the similarity and the difference between the regions and ‘cultural difference’?

It lacks proper description of the findings E.g., Use of figure.

Strengthen the discussion section.

The conclusion and the recommendation should be in line with the findings of the study. Likewise, specific recommendation should be conveyed.

The manuscript didn’t declare ‘conflict of interest’

Regards,

Reviewer #3: First of all, I would like to thank the editor for inviting me to review this manuscript. Then, the authors for coming with an important topic.

Abstract

The document needs intensive revision for grammatical issues and sentence organization including use of tenses and conjugations in the appropriate place, and punctuations.

Suggest to modify the first sentence. To make it sound, don’t start with even though.

Methods: who did cross-sectional study? It is not your data, please refer the source of the data then state how they capture it. Otherwise, it seems you have actively participated in the data collection process.

Suggest to replace “prevalence” by other terms for explaining full antenatal utilization.

Report CI for rural and urban ANC utilization.

Rural ANC utilization was higher than urban. Your explanation have not supported your findings. ” So to narrow this gap, emphasis should be given to both resource distribution targeting to rural households, improvement of maternal education and creating a plat form to access information about the service and its relevance”. Please provide evidence based justification and recommend based on your findings.

Introduction

Line 52 and 53: suggest to modify the sentence

Line 54: use updated reference

Elaborate the introduction starting from extent of poor ANC utilization, its effect on maternal and newborn health, factors affecting ANC utilization and the observed disparities by country, region, and residence. Consequently, explain the interventions posed before for improving ANC utilization and gap of previous studies for showing disparities. Clearly set the gap. Keep flow of the sentences.

What is the relevance of studying urban and rural disparities? Reference 22, 14, and 15 also studied disparities in ANC utilization in Ethiopia. What values you added?

Line 75: “Because though” please revise the document for English grammar.

Methods and Materials

Line 92: better to report study population instead of sampled population.

If a mother provides two or three live births within five years, what measures are taken during sampling? Which ANC was taken? If there is disparities in ANC utilization for different pregnancies within a mother, in which category you classified this particular participant? This all needs to be clarified.

Results

Suggest to provide clear interpretation for your decomposition results. “That is, if urban women distribution of primary, secondary and higher education shifted to rural levels the rural-urban disparity would increase by 3.76%, 11.84%, and 15.27% respectively”

Suggest to describe about difference due to characteristics and difference due to coefficients in methods section.

Discussion

Inconsistent result is reported in result and discussion section. You reported “The disparity of full antenatal care utilization between rural and urban areas was high (rural prevalence was 27.73%, while in urban areas it was 15.52%)” , whereas in result section you reported “The result of this study showed that full antenatal care utilization among women in urban areas of Ethiopia was better than that of rural residents” in discussion.

Did you consider changes in characteristics (endowment) and change in behavior (coefficient) as a variable?

What is your base for discussing “changes in characteristics (endowment) than change in behavior (coefficient)”?

Line 227 to 229: suggest to provide evidence based justification for the particular finding. How you relate Muslim religion and ANC utilization. Here, you have to provide justification for the disparities occurred between urban and rural areas.

Add practical implication for pertinent findings.

Your Interpretations are not still clear in discussion. Please provide clear and simple interpretations.

Line 242 to 244. “This result is in line with studies conducted in Ethiopia (13, 14, 22),” if ample evidences are available in Ethiopia, what you have added???? You have used almost all references in your discussion from Ethiopia.

Line 256: add reference for your justification.

Reviewer #4: Question to be answered

1. What basic observational disparities the authors observed to do this decomposistion analysis inEthiopia?

2. Why not the authors accounted the effect of endowment and coeffient interaction explanation on the gaps of ANC utilization?

3. There are need of presentation of some findings further by tables/graphs, but missed. For example, line numbers 192 and 205

4. What will be the authors scientific recommendation for the covariate religion explanation for ANC urban-rural disparity? I recommend you to omit.

5. Why the authors lack similar findings of religious difference? Can you declare that it is a novel finding?

The overall comments and some questions are highlighted in the manuscript.

I also attached a separate file.

Reviewer #5: There are some places that need to be grammatically checked. I noticed that there are some places that are missing spaces. One such instance is between a word and the reference number. There are places that have spaces and some that don't. Another discrepancy I see is that you put commas for numbers greater than or equal to 1,000 (not referencing a year). Please make sure the spacing between the references follow the PLOS submission guideline.

7. PLOS authors have the option to publish the peer review history of their article (what does this mean?). If published, this will include your full peer review and any attached files.

Reviewer #2: No

Reviewer #3: No

Reviewer #4: **Yes: **Wolde Melese Ayele

Reviewer #5: No

---

## [Author Response · Author response to Decision Letter 1]

9 Aug 2024

We are very thankful for your constructive comments and we learn a lot through out revising the document.

---

## [Editor Report · Decision Letter 2]

9 Sep 2024

Rural-urban disparities in fullantenatal care utilization among women in Ethiopia: A further Analysis of Mini-EDHS, 2019

PONE-D-23-21294R2

Dear Dr. Adissu,

We’re pleased to inform you that your manuscript has been judged scientifically suitable for publication and will be formally accepted for publication once it meets all outstanding technical requirements.

Kind regards,

Doris Verónica Ortega-Altamirano, PhD

Academic Editor

PLOS ONE

Additional Editor Comments (optional):

I am pleased to approve the version of manuscript PONE-D-2321294-R2.

The manuscript presents a useful study on understanding the phenomenon of using full antenatal care in women of reproductive age in Ethiopia.
---

## [Editor Report · Acceptance letter]

10 Oct 2024

PONE-D-23-21294R2 

PLOS ONE

Dear Dr. Addisu, 

I'm pleased to inform you that your manuscript has been deemed suitable for publication in PLOS ONE. Congratulations! Your manuscript is now being handed over to our production team.

Kind regards, 

on behalf of

Dr. Doris Verónica Ortega-Altamirano 

Academic Editor

PLOS ONE